# Both Autopsy and Computed Tomography Are Necessary for Accurately Detecting Rib Fractures Due to Cardiopulmonary Resuscitation

**DOI:** 10.3390/diagnostics10090697

**Published:** 2020-09-15

**Authors:** Kunio Hamanaka, Kei Nishiyama, Mami Nakamura, Marin Takaso, Masahito Hitosugi

**Affiliations:** 1Department of Legal Medicine, Shiga University of Medical Science, Otsu 520-2192, Japan; mamin@belle.shiga-med.ac.jp (M.N.); marint@belle.shiga-med.ac.jp (M.T.); hitosugi@belle.shiga-med.ac.jp (M.H.); 2Department of Trauma and Critical Care, Kyoto Medical Center, Kyoto 612-8555, Japan; keinishi@kuhp.kyoto-u.ac.jp

**Keywords:** autopsy, computed tomography, CPR, post-mortem, rib fracture

## Abstract

Few studies have compared the sensitivities of autopsy and post mortem computed tomography (PMCT) in detecting rib fractures caused by cardiopulmonary resuscitation (CPR). We aimed to compare the characteristics between both modalities for accurately detecting CPR-related rib fractures. This single-centre observational study included adult patients with autopsy records and PMCT scans at our institution from January 2013 to March 2019. CPR-related rib fractures were evaluated using autopsy and PMCT findings. In 62 patients enrolled, 339 rib fractures were detected on autopsy and/or PMCT (222 fractures on both PMCT and autopsy, 69 on PMCT alone, and 50 on autopsy alone). The agreement of detection for both modalities was substantial (kappa coefficient, 0.78). In the logistic regression model, incomplete fractures detected by PMCT and age <75 years were significantly associated with findings that were negative on autopsy but positive on PMCT, while rib number (ribs 1–3 and 7–12) and fracture location (posterolateral and paravertebral) were significantly associated with negative PMCT findings but positive autopsy findings. Autopsy and PMCT showed complementary roles, and are thus necessary in accurately detecting CPR-related rib fractures. Combining both modalities may contribute to improved CPR quality and better understanding of discrepancy in characteristics between the two modalities.

## 1. Introduction

High-quality chest compressions are the most important component of cardiopulmonary resuscitation (CPR) and is essential to improving survival [1]. However, chest compressions during CPR can cause various iatrogenic injuries to the patient. According to several studies, the most common complication of CPR is rib fracture (13–97%) [2,3,4,5]. Multiple rib fractures can cause haemothorax, pneumothorax, or intrathoracic visceral injuries, which can negatively affect the quality of life (QOL) of the patient.

Autopsies are considered the gold standard for detecting CPR-related injuries. However, recently, post mortem computed tomography (PMCT) has been considered as an alternative [6]. Few studies have compared the sensitivities of autopsy and PMCT in detecting CPR-related rib fractures [7,8,9]. Two of the studies revealed that autopsies are more sensitive than PMCT in detecting CPR-related rib fractures [7,9], but the other study showed an opposite result [8]. Although a relatively low rate of congruence between these modalities has been reported, the reason for such disagreement in detecting CPR-related rib fractures is not fully understood.

If both modalities differ in characteristics in detecting CPR-related rib fractures, then both can be used for a more accurate detection. Subsequently, the use of both modalities may contribute to improvement of the quality of CPR, and to better understanding the discrepancies in characteristics between these modalities. This study therefore aimed to evaluate and compare the characteristics of autopsy and PMCT in detecting CPR-related rib fractures, by analysing data of patients who underwent these procedures.

## 2. Materials and Methods

### 2.1. Study Design

This is a single-centre retrospective observational study. We included patients examined by autopsy and PMCT at the Shiga University of Medical Science from January 2013 to March 2019. The exclusion criteria were as follows: (1) lack of whole-body PMCT scans, (2) traumatic cardiac arrest, (3) age under 18 years, or (4) CPR was not performed. The study protocol was approved by the institutional review board of the Shiga University of Medical Science (R2015-127) on 27 October 2015.

### 2.2. Data Collection

We collected data on age, sex, height, weight, and cause of death of the patients. We also collected data on rib fractures (rib number, thoracic side, and location of rib fracture) from the autopsy records and data on rib fractures (rib number, thoracic side, and location and type of rib fracture) from the PMCT images. Rib fractures were classified into 76 parts, as described in previous studies [10,11] and grouped into four locations, as follows: parasternal, anterolateral, posterolateral, and paravertebral. The anterolateral area was defined as that area within the midclavicular and ventral axillary lines, and the posterolateral area as that within the midscapular and dorsal axillary lines.

### 2.3. PMCT

Every hospital in Shiga Prefecture performs PMCT immediately after a patient is pronounced dead. In this study, axial whole-body PMCT images of 5 mm slice thickness were analysed by a board-certified clinical radiologist who was blinded to the autopsy results. The location and type of the rib fracture were determined. Rib fracture was classified as complete fracture, outer cortical fracture (fracture of the external cortical plate), inner cortical fracture (fracture of the internal cortical plate), or buckle fracture, according to a recent PMCT study (Figure 1) [11]. A buckle fracture was characterised by a smooth outer cortex and kink-like fracture to the inner cortex.

### 2.4. Forensic Autopsy

Forensic autopsies were performed by board-certified forensic pathologists who were blinded to the PMCT results. The ribs were routinely examined as part of a standard autopsy protocol. After a midline incision, the subcutaneous and thoracic organs and tissues were assessed for haemorrhages and structural stability. The thoracic cage was opened in a conventional manner, but with great care to avoid modifying the already-detected findings. The cavity side of the thoracic cage was examined for visible fissures/fractures and misalignment of the ribs and sternum [8].

### 2.5. Statistical Analysis

Data were expressed as medians and interquartile ranges. To define the agreement of detection between autopsy and PMCT, the mean Cohen’s kappa coefficient was calculated. A kappa coefficient of +1 indicated perfect agreement, 0 indicated an agreement no better than chance, and −1 represented absolute disagreement. The chi-squared test was used to compare the prevalence of a fracture type between the PMCT and autopsy findings. To determine important factors for each disagreement, such as the presence of findings that were negative on autopsy but positive on PMCT or findings that were negative on PMCT but positive on autopsy, adjusted odds ratios were calculated using multiple logistic regression analysis. A two-tailed *p*-value of <0.05 was considered statistically significant. Statistical analysis was carried out using JMP software version 10.0.0 (SAS Institute, Cary, NC, USA).

## 3. Results

### 3.1. Patient Characteristics

During the study period, data on 158 patients with forensic autopsy and PMCT findings were collected. The following patients were excluded: two patients with no whole-body PMCT scans, 70 patients with a traumatic cardiac arrest, 18 patients aged <18 years, and six patients who did not undergo CPR. Thus, this study enrolled a total of 62 patients, of whom 39 patients were male and 23 were female. Among them, 40 patients had rib fractures and 22 did not have rib fractures. The general characteristics of the patients are listed in Table 1.

### 3.2. Characteristics of Rib Fractures

In all 62 patients, we examined 4712 bone components by autopsy records and PMCT scans. A total of 339 rib fractures were detected by autopsy and/or PMCT. Of the 339 fractures, 289 were detected by PMCT and 272 by autopsy. A total of 222 rib fractures were detected by both PMCT and autopsy, 69 by PMCT alone, and 50 by autopsy alone. The kappa value was 0.78, which indicated substantial agreement.

The most common fracture site was Rib 4 (72 fractures), and the number of rib fractures decreased with the upper and lower ribs. PMCT detected more fractures than did autopsy in ribs 4–6, whereas autopsy detected more fractures than did PMCT in other ribs. The kappa value showed an almost perfect agreement for Rib 2, 5, and 6 fractures, substantial agreement for Rib 3, 4, and 7 fractures, moderate agreement for Rib 8 and 9 fractures, and no agreement for Rib 10 fractures (Table 2).

The number of fractures per area was as follows: parasternal, 171; anterolateral, 151; posterolateral, 8; and paravertebral, 9. PMCT detected more fractures than did autopsy in the parasternal and anterolateral areas, while autopsy detected more fractures than did PMCT in the paravertebral area. The kappa value showed substantial agreement for the parasternal and anterolateral areas, and moderate agreement for the posterolateral and paravertebral areas (Table 2).

The analysis of rib fractures according to the thoracic side and patient’s sex, age, weight, and body mass index yielded a kappa value of 0.76–0.82 (Table 2).

### 3.3. Detection by Autopsy according to Fracture Type on PMCT

For fractures detected by PMCT, the number of fractures by fracture type was as follows: complete fractures, 106; outer cortical fractures, 14; inner cortical fractures, 22; and buckle fractures, 146. Agreement of PMCT findings with autopsy findings was highest for complete fractures and lowest for buckle fractures. Thus, agreement of PMCT findings with autopsy findings was significantly dependent on the type of fracture detected by PMCT (Table 3).

### 3.4. Logistic Regression Model for Findings That Were Negative on Autopsy but Positive on PMCT

The type of fracture detected by PMCT and age were significantly associated with findings that were negative on autopsy but positive on PMCT. It was more difficult to detect incomplete fractures (outer cortical, inner cortical, and buckle) on autopsy than on PMCT (odds ratio, 8.49; *p* < 0.01). It was also more difficult to detect fractures in patients aged <75 years on autopsy than on PMCT (odds ratio, 3.83; *p* < 0.01). Meanwhile, the rib number, fracture location, and sex, height, and weight of the patient were not significantly associated with findings that were negative on autopsy but positive on PMCT (Table 4A).

### 3.5. Logistic Regression Model for Findings That Were Negative on PMCT but Positive on Autopsy

The rib number and location of rib with fracture were significantly associated with findings that were negative on PMCT but positive on autopsy. For ribs 1–3 and 7–12, it was more difficult to detect fractures on PMCT than on autopsy (odds ratio, 3.07; *p* < 0.01). It was also more difficult to detect fractures in the posterior areas (posterolateral and paravertebral) on PMCT than on autopsy (odds ratio, 4.76; *p* = 0.01). Sex, height, weight, and age were not significantly associated with findings that were negative on PMCT but positive on autopsy (Table 4B).

## 4. Discussion

To our knowledge, this is the first study to evaluate in detail the disagreement of detecting CPR-related rib fractures between autopsy and PMCT. Although the mean kappa coefficient for our findings was 0.78, which is not an almost perfect agreement, it indicated substantial agreement. Therefore, PMCT and autopsy had complementary roles in detecting rib fractures caused by CPR.

We found that the agreement between PMCT and autopsy findings was significantly associated with the type of rib fracture detected on PMCT, and such agreement was lowest for buckle fractures. *Buckle* is an engineering term used to describe a fracture occurring on the compressed or inner side of a linear structure (e.g., a rib) without wholly disrupting the tensile side or outer surface of the structure [12]. Buckle rib fractures are more difficult to detect on plain radiographic examination and may be hard to clearly define, even on autopsy [12]. However, so far, no studies have compared the sensitivities of PMCT and autopsy in detecting buckle rib fractures, except our study.

In this study, incomplete fractures, especially buckle fractures, were associated with findings that were negative on autopsy but positive on PMCT. Half of the fractures detected by PMCT were diagnosed as buckle fractures. This study therefore proved that buckle fractures are difficult to detect on autopsy. On the other hand, we found good agreement between PMCT and autopsy findings for complete fractures. These findings indicate that detecting fractures on autopsy mainly depends on stability testing. As incomplete fractures are considered to be more stable and undetectable than other fractures, we suggest PMCT, rather than autopsy, as the gold standard for detecting incomplete fractures.

We also found that an age of <75 years was associated with findings that were negative on autopsy but positive on PMCT. The reason for this could be as follows. As age increases, the bone mineral density of the ribs decreases, the ribs become more brittle, and the rib shape becomes more elongated and flatter [13]. Meanwhile, younger patients have more flexible ribs. Therefore, it might be more difficult to evaluate instability on autopsy for younger patients with rib fractures due to bone flexibility.

This study showed that rib number (i.e., ribs 1–3 and 7–12) and fracture location (i.e., posterolateral and paravertebral areas) were associated with findings that were negative on PMCT but positive on autopsy. The anterior area and ribs 4–6 are more prone to direct force from chest compressions, so dislocation with fracture in these locations is likely to occur. In the posterior area, especially at ribs 1–3 and 7–12, fractures may likely occur owing to an indirect force, though dislocations are likely to be slight even if there are fractures with instability. It may be difficult to detect rib fractures without a dislocation on PMCT, even rib fractures with cortical rupture and instability. Therefore, we strongly suggest that neither autopsy nor PMCT alone is the gold standard for detecting CPR-related injuries. Conversely, combining autopsy and PMCT could help in diagnosing rib fractures more accurately.

Previous studies have shown that conventional chest radiographs provide a limited view of rib fractures and that autopsy by a trained forensic pathologist was a more accurate method of detecting rib fractures [14]. Autopsy has also been reported to be more sensitive than PMCT in detecting rib fractures. However, PMCT provides a fast and high-resolution view of the skeleton and has recently been considered as an alternative to autopsy. In this study, we detected more rib fractures on PMCT than on autopsy. All autopsies in this study were performed to investigate the cause of cardiac arrest, not for analysing the complications of CPR. Furthermore, as CPR-related rib fractures are accompanied by less haemorrhage around the fractured site, they might not be more accurately detected. On the other hand, in this study, PMCT was specifically performed for detecting and analysing rib fractures, with the radiologist being blinded to the autopsy results. These findings further support our notion that neither autopsy nor PMCT alone is the gold standard for detecting CPR-related rib fractures and that combining both modalities is necessary for accurately diagnosing rib fractures in the future.

Chest compressions are a key component of effective CPR [1]. Chest compressions with a depth of approximately 5 cm were associated with a greater likelihood of favourable outcomes than shallower compressions [15]. The 2010 American Heart Association (AHA) Guidelines for CPR recommended a target chest compression depth of at least 5 cm (2 inches) [1]. However, chest compression-related injuries were more common when the compression depth was >6 cm (2.4 inches) than when it was between 5 and 6 cm [16]. The 2015 AHA Guidelines for CPR updated the chest compression depth to between 5 and 6 cm (2.4 inches) [1]. These data imply that it is important to evaluate rib fractures repeatedly and apply the results in the clinical setting to avoid CPR complications. Therefore, more accurate methods to detect rib fractures, such as combining autopsy and PMCT, may be beneficial in this regard.

This study has some limitations. First, the diagnostic criteria for rib fractures were identified by either autopsy or PMCT, or both, because there is no gold standard yet for detecting CPR-related rib fractures. This situation might have biased the findings as some rib fractures may not be detected by dissection or PMCT. Second, although PMCT is being performed in every hospital in Shiga Prefecture, there is no standard protocol for performing such procedure. Lastly, the PMCT findings were retrospectively interpreted by only one radiologist, so inter-observer variation could not be calculated. A previous study determined that identification of rib fractures by readers is not entirely consistent [11]. Therefore, in this study, depending on the radiologist’s experience, there may have been over- or under-diagnosis of rib fractures.

## 5. Conclusions

This study revealed that autopsy and PMCT have complementary roles, and are thus necessary in accurately detecting rib fractures caused by CPR. We also found that incomplete fractures and patient age of <75 years were associated with findings that were negative on autopsy but positive on PMCT, while rib number (ribs 1–3 and 7–12) and fracture location (posterolateral and paravertebral areas) were associated with negative findings on PMCT but positive findings on autopsy. Combining autopsy and PMCT may contribute to improved quality of CPR and to a better understanding of the discrepancy in characteristics between these two modalities.

## Figures and Tables

**Figure 1 diagnostics-10-00697-f001:**
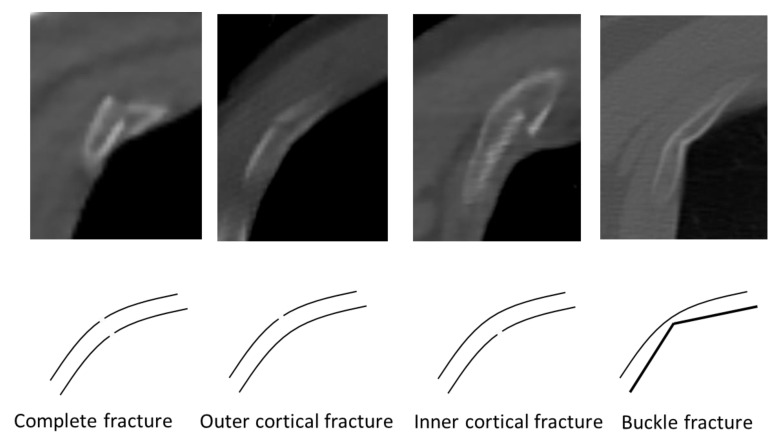
Type of rib fracture detected on post mortem computed tomography. A complete fracture was documented when both cortical lines were disrupted. A single dehiscence of either the outer or inner cortical line and the formation of a sharp angle at the inner cortical line called “buckle fracture” were individually documented and summarised as incomplete fractures.

**Table 1 diagnostics-10-00697-t001:** Demographic characteristics of patients.

Variable	*n* = 62
Age (years)	60 (42–77)
Male sex	39 (63%)
Height (cm)	162 (156–172)
Weight (kg)	54 (45–69)
Body mass index (kg/m^2^)	21 (18–24)
Patients with rib fractures	40 (65%)
No. of rib fractures per patient	4 (0–11)
Cause of death	
Cardiac origin	24 (39%)
Respiratory disease	11 (18%)
Cerebral disorder	8 (13%)
Other	19 (31%)

Data are presented as number (percentage) or mean/median (interquartile range).

**Table 2 diagnostics-10-00697-t002:** Characteristics of rib fractures.

Variables	All Detected Fractures	Detected by PMCT	Detected by Autopsy	Kappa Value
**At all ribs**	339	289	272	0.78
**Rib number**				
1	7	0	7	N/A
2	40	34	36	0.85
3	60	54	47	0.79
4	72	66	55	0.78
5	65	60	52	0.82
6	48	44	41	0.86
7	31	22	23	0.60
8	10	5	8	0.45
9	4	3	2	0.39
10	2	1	1	0.00
11	0	0	0	N/A
12	0	0	0	N/A
**Fracture location**				
Parasternal	171	144	136	0.74
Anterolateral	151	136	122	0.80
Posterolateral	8	5	5	0.40
Paravertebral	9	4	9	0.61
**Thoracic side**				
Right	184	157	149	0.78
Left	155	132	123	0.77
**Sex**				
Female	165	134	138	0.77
Male	174	155	134	0.79
**Height**				
<160 cm	195	161	164	0.78
≥160 cm	144	128	108	0.77
**Weight**				
<50 kg	212	177	177	0.78
≥50 kg	127	112	95	0.76
**BMI**				
<21 kg/m^2^	185	160	148	0.79
≥21 kg/m^2^	154	129	124	0.77
**Age**				
<75 years	148	131	101	0.71
>75 years	191	158	171	0.82

Data are presented as numbers. Kappa values were determined by Bowker analysis. BMI, body mass index; N/A, not applicable; PMCT, post mortem computed tomography.

**Table 3 diagnostics-10-00697-t003:** Types of rib fractures detected on PMCT and corresponding autopsy findings.

Type of Fracture on PMCT	Detected on Autopsy, n (%)	*p*-Value
Complete (*n* = 106)	100 (94)	<0.01
Outer cortical (*n* = 14)	9 (64)	
Inner cortical (*n* = 23)	22 (96)	
Buckle (*n* = 146)	91 (62)	

*p*-values were determined by chi-squared test. PMCT, post mortem computed tomography.

**Table 4 diagnostics-10-00697-t004:** (A). Logistic regression model for findings negative on autopsy but positive on PMCT. (B). Logistic regression model for findings negative on PMCT but positive on autopsy.

**A**		
**Parameter**	**Odds Ratio**	***p*-Value**
Rib number		
4–6	0.88	0.67
1–3 and 7–12	1.14	
Location		
Anterior (parasternal, anterolateral)	0.23	0.09
Posterior (posteolateral, paravertebral)	4.33	
Type		
Complete fractures	0.12	<0.01*
Incomplete fractures (outer cortical, inner cortical, and buckle)	8.49	
Sex		
Female	1.13	0.73
Male	0.88	
Height		
<160 cm	0.91	0.82
≥160 cm	1.10	
Weight		
<50 kg	0.67	0.33
≥50 kg	1.50	
Age		
<75 years	3.83	<0.01*
≥75 years	0.26	
**B**		
**Parameter**	**Odds Ratio**	***p*-Value**
Rib number		
4–6	0.33	<0.01*
1–3 and 7–12	3.07	
Location		
Anterior (parasternal, anterolateral)	0.21	0.01*
Posterior (posteolateral, paravertebral)	4.76	
Sex		
Female	1.74	0.15
Male	0.57	
Height		
<160 cm	1.13	0.78
≥160 cm	0.88	
Weight		
<50 kg	0.86	0.74
≥50 kg	1.16	
Age		
<75 years	1.42	0.36
≥75 years	0.70	

* Statistically significant, *p* < 0.05. PMCT, post mortem computed tomography.

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
