# Peer review of "Both Autopsy and Computed Tomography Are Necessary for Accurately Detecting Rib Fractures Due to Cardiopulmonary Resuscitation"

_diagnostics, 2020, doi:10.3390/diagnostics10090697_

Round 1

Reviewer 1 Report

The paper aimed to evaluate and compare the characteristics of autopsy and PMCT in detecting CPR-related rib fractures, by analyzing data of patients who underwent these procedures.
The idea is interesting and the work is done with correct methodology.
I have only minor observations to complete the paper.
1. Page 2, line 61, correct: "studies, 10,11 and grouped into four locations as follows: parasternal, anterolateral, posterolateral ...".
2. When the authors show the schematic images of the PMCT fractures (Figure 1) they should then add, in the appropriate paragraph, a photo consisting of 4 images that allow to compare the fractures at the autopsy table. It is particularly important to show Buckle fractures.
3. Indeed, as the authors point out, an important limitation of the paper is the fact that the PMCT findings were retrospectively interpreted by only one radiologist, so inter-observer variation could not be calculated. So, I wonder: have the authors noticed any diagnostic excesses and / or differences among individual radiologists in diagnosing the type and location of fractures? This point could be interesting to bring out difficulties with PMCT by less experienced radiologists.

Reviewer 2 Report

The authors present a comparison study between PMCT and autopsy: the ribs fractures post resuscitation are the object of the research. The study is well done and statistical calculation are correct. The results are similar to those of other similar studies (correctly cited in the references). The complementarity of PMCT and autopsy should be more emphasized: in fact, in these studies, the quality of the autopsy is always a topic that has not objective tools of analysis.
